# Graph Neural Network-Based Efficient Subgraph Embedding Method for Link Prediction in Mobile Edge Computing

**DOI:** 10.3390/s23104936

**Published:** 2023-05-20

**Authors:** Xiaolong Deng, Jufeng Sun, Junwen Lu

**Affiliations:** 1School of Cyberspace Security, Key Laboratory of Trustworthy Distributed Computing and Service, Ministry of Education, Beijing University of Posts and Telecommunications, Beijing 100876, China; 2School of Computer and Information Engineering, Xiamen University of Technology, Xiamen 361024, China; jwlu@xmut.edu.cn; 3School of Cyberspace Security, Beijing University of Posts and Telecommunications, Beijing 100876, China; sunjufeng@bupt.edu.cn

**Keywords:** link prediction, graph neural network, graph embedding, 5G MEC network routing links

## Abstract

Link prediction is critical to completing the missing links in a network or to predicting the generation of new links according to current network structure information, which is vital for analyzing the evolution of a network, such as the logical architecture construction of MEC (mobile edge computing) routing links of a 5G/6G access network. Link prediction can provide throughput guidance for MEC and select appropriate c nodes through the MEC routing links of 5G/6G access networks. Traditional link prediction algorithms are always based on node similarity, which needs predefined similarity functions, is highly hypothetical and can only be applied to specific network structures without generality. To solve this problem, this paper proposes a new efficient link prediction algorithm PLAS (predicting links by analysis subgraph) and its GNN (graph neural network) version PLGAT (predicting links by graph attention networks) based on the target node pair subgraph. In order to automatically learn the graph structure characteristics, the algorithm first extracts the h-hop subgraph of the target node pair, and then predicts whether the target node pair will be linked according to the subgraph. Experiments on eleven real datasets show that our proposed link prediction algorithm is suitable for various network structures and is superior to other link prediction algorithms, especially in some 5G MEC Access networks datasets with higher AUC (area under curve) values.

## 1. Introduction

In real life, many complex systems can be modeled into complex networks for analysis, such as power networks, traffic networks, routing networks, citation networks, 5G/6G space–air–ground communication networks, social networks, etc.

In link prediction tasks, nodes in the network always represent real entities, such as routers and switches in the network, and associations between entities represent edges. Link prediction is mainly based on current network structures and other information to complete the missing links in a current network or predict the possible new connections in a future network [1], such as new routing links, which may occur in the 5G/6G space–air–ground communication networks around satellite MEC equipment (mobile edge computing) to transport data-dense computation. As an important research direction in complex networks, link prediction has extensive theoretical research value and practical application value.

In other theoretical research value aspects, link prediction can reveal the evolutionary mechanism of a network and provide a simple and fair comparison method for the evolutionary network [2]. For example, for a certain type of network, many models provide evolutionary methods. The quality of evolutionary methods can be verified through real datasets, but these evolutionary methods are often limited by the scale of evolutionary time or the difficulty of real dataset collection. The link prediction algorithm provides a simple and fair comparison method. According to network structure at the moment t−n, the link prediction algorithm can be used to complete the missing link in the current network or to predict the network at time t when new links are generated in the future. Then, the accuracy of different evolution methods can be obtained via comparison with the network at the original moment. Additionally, the advantages and disadvantages of different evolutionary methods can be obtained by analyzing the accuracy of new generated links.

In other practical applications, for example, in social networks, the link prediction algorithm recommends users who have the same interests but are not friends to other online social users to improve user stickiness [3]. In Weibo or Facebook, the link prediction algorithm can be used to recommend topics or short videos that users are interested in and to predict the popularity of certain special topics and short videos [3]. In the field of e-commerce, the relationship graph between users and commodities can be established, and the link prediction algorithm can recommend relevant commodities to users, reduce the time of searching for commodities, and improve efficiency [4]. In the protein network, there are many unknown links, and searching for these links requires many crossover experiments, which will waste much manpower and material resources. However, the link prediction algorithm will predict the most likely links, which provides guiding opinions for the experiment, shortens the scope of the experiment, and speeds up the identification process of unknown links. Additionally, the experimental cost is reduced [5]. The research on link prediction in complex networks has wide theoretical and practical value.

## 2. Related Work

In reality, a large number of complex systems can be represented by networks, with nodes representing different entities and edges representing relationships between entities. Link prediction is the prediction of missing links in the current network and the generation of new links in a future network through the known network structure and other information [1]. It has a wide range of practical value regarding friend recommendation [3], product recommendation [4], knowledge graph completion [6], and other areas [7,8,9].

The heuristic algorithms commonly used in link prediction are based on node similarity [10]. This algorithm assigns a scoring function to each node pair, representing the similarity of node pair, and sorts the unobserved node pairs according to the scoring function. The node pairs with high similarity are more likely to generate new connections. This kind of algorithm can be classified according to the maximum hop number of the maximum neighbor node required to calculate the scoring function [11]. For example, CN [12] and JC [13] link prediction algorithms only need the one-hop neighbor node of the target node pair to calculate the score function, so it belongs to the heuristic algorithm of one-hop node similarity. AA [3] and RA [14] link prediction algorithms need the target node to calculate the two-hop neighbor node when calculating the score function. Therefore, it belongs to the heuristic algorithm of two-hop node similarity. This form of similarity-based heuristic algorithm has become the most common algorithm in link prediction because of its simplicity and effectiveness. However, such algorithms need strong assumptions. When changing from one network structure to another network structure, the assumptions are not consistent with the other network structure. For example, the CN algorithm believes that the more common neighbors two nodes have, the more likely they will have links in the future, which is often correct in social networks. However, this is not true in protein interaction networks (the more common neighbors two nodes have, the less likely they are to generate links in the future) [15]. Therefore, it is a significant disadvantage of heuristic algorithms based on node similarity to select appropriate scoring functions for different network structures.

The link prediction algorithm based on machine learning mainly transforms the link prediction task into a binary classification task, in which the node pairs with links are regarded as positive classes and the node pairs without links are regarded as negative classes. The key to this kind of algorithm mainly lies in the selection of features and classification algorithms. In 2004, Faloutsos et al. [16] introduced a connection subgraph, which is able to capture the topology between two nodes in a social network. In 2006, Al et al. [17] extracted the non-topological features of the network based on the extraction of the topological features of the network, which improved the algorithm’s accuracy. In 2007, Liben et al. [18] extracted some network topology features from the citation network, such as CN, AA, Katz, etc., and input them into a supervised learning algorithm for learning and prediction. Their experimental results outperform link prediction algorithms based on individual network topologies. In 2010, Benchettara et al. [19] used the enhanced decision tree algorithm and found that using topological features in the feature set is able to significantly improve the link prediction algorithm’s precision, recall, and F value. In 2014, Fire et al. [20] proposed a set of easily computable graph-structured features and adopted two ensemble learning methods to predict missing links in the network. In 2018, Zhang et al. [21] used the attribute features of nodes as non-topological features to input into supervised learning algorithms, improving the accuracy of the link prediction algorithms. In 2018, Mandal et al. [22] used a variety of supervised learning algorithms for link prediction in a two-layer network composed of Twitter and Foursquare. In 2022, Kumar et al. [23] introduced the value of node centrality as a sample feature, input it into various supervised learning algorithms for prediction, and achieved the best results on the LGBM (light gradient boosted machine) classifier. The key to link prediction algorithms based on machine learning lies in selecting feature sets. Such algorithms often extract some topological features of the network as feature sets. However, when solving domain-specific link prediction problems, corresponding domain knowledge is also required to construct its domain-specific features. Compared with the heuristic-based link prediction algorithm, the link prediction algorithm based on machine learning can improve its accuracy, but it also results in time costs for training models and feature selection.

Link prediction algorithms based on graph representation learning mainly map high-dimensional dense matrices (graph data) into low-dimensional dense vectors and then use the mapped vectors for downstream tasks, such as node classification [24,25], graph classification [26,27], link prediction [28], etc. In 2014, Perozzi et al. [29] first proposed the graph representation algorithm DeepWalk for unsupervised learning. The algorithm obtains the sequence of nodes through random walks and inputs the sequence of nodes as sentences into the Skip-Gram model in the Word2Vec algorithm to obtain the node vector representation. In 2015, Tang et al. [30] proposed the LINE algorithm model, which proposed a method of edge sampling so that the vectors of the obtained nodes retain the first-order similarity and second-order similarity. In 2016, Grover et al. [31] proposed the Node2Vec algorithm. Node2Vec is similar to DeepWalk, but Node2Vec uses a biased random walk method, which balances depth-first walk and breadth-first walk and obtains a higher-quality embedded representation. In 2016, Wang et al. [32] proposed the SDNE (structural deep network embedding) model. SDNE is a semi-supervised deep learning model that uses a deep network structure to simultaneously optimize the first-order and second-order similarity objective functions and obtains vectors for preserving the graph’s local and global structure. In 2016, Cao et al. [33] proposed the DNGR algorithm model. DNGR uses the random walk model (random surfing) to generate the probability co-occurrence matrix, calculates the PPMI matrix with the probability co-occurrence matrix, and uses the superimposed denoising automatic encoding machine to extract features to obtain the vector representation of the node. In 2016, Kipf et al. [34] proposed the GCN (graph convolutional network) model. The algorithm considers the influence of neighbor nodes and continuously aggregates the characteristics of neighbor nodes. Embedding neighbor nodes can obtain scalability, and the global information can be described by aggregating the characteristics of neighbor nodes through multiple iterations. In 2016, Kipf et al. [35] proposed the VGAE (variational graph auto-encoders) model, introducing variational autoencoders into graph data. The distribution of the node vector representation of the known graph is learned through GCN convolution, the representation of the node vector is sampled in the distribution and then decoded (link prediction) to reconstruct the graph. In 2017, Veličković et al. [36] proposed the GAT (graph attention networks) model, which introduced an attention mechanism. When calculating the vector representation of nodes, the model’s generalization ability is improved by assigning different weights to the characteristics of nodes. At the same time, a multi-head attention mechanism is introduced and the features obtained by multiple attention mechanisms are spliced and averaged to obtain the final node representation. In 2017, Hamilton et al. [37] proposed the GraphSage algorithm model for large-scale graph data. By learning an aggregation function, the neighbor nodes are sampled and aggregated to obtain a new vector representation of the node. In 2018, Chen et al. [38] proposed the HARP algorithm model. HARP selects the starting node by weight and combines it with DeepWalk and Node2Vec to obtain a better embedding representation. In 2018, Schlichtkrull et al. [39] proposed the R-GCN (relational graph convolutional networks) algorithm model. R-GCN introduced weight sharing and parameter constraints to improve the performance of the link prediction algorithm. In 2018, Sam et al. [40] learned the representation vectors of nodes in historical time slices through the Node2Vec algorithm, concatenated the vectors of nodes in historical time slices to obtain the representation of future time slice links, and finally used supervised learning algorithms to predict future time links state. In 2019, Lei et al. [41] used GCN to learn the network topology features of each time slice, used LSTM (long short term memory) to capture the evolution pattern of multiple continuous time slice dynamic networks, and then used GAN (generative adversarial network) to generate a representation of links in future time slices. In 2017, Zhang et al. [42] proposed the WLNM link prediction model, which extracts the h-hop closed subgraph of the target node pair and sends the adjacency matrix of the subgraph to the fully connected layer for learning, which improves the link prediction algorithm results. Link prediction methods based on graph representation learning capture deeper network structural relationships in complex networks and more complex relationship features between nodes with difficulty due to limited walk steps and aggregation methods, resulting in lower algorithm accuracy.

Considering that the existing link prediction algorithms are unsuitable for different network structures, capturing the deeper network structure relationships and the more complex relationship characteristics between nodes is difficult, this paper proposes a link prediction algorithm based on subgraph (PLAS, predicting links by analysis subgraph). The algorithm first obtains h-hop neighbor nodes of the target node pair to form a subgraph and then assigns labels to each node of the subgraph. The nodes in the subgraph are sorted according to labels. Finally, the nodes of the subgraph are input to the full connection layer in a consistent order for classification.

Additionally, our contributions are as follows:A subgraph node labeling method is provided, which is able to automatically learn graph structure features and input nodes of subgraphs into the full connection layer in a consistent order.A link prediction method (PLAS) based on subgraph is proposed, which can be applied to different network structures and is superior to other link prediction algorithms.Based on a torch, the link prediction algorithm (PLAS) model based on subgraph is implemented and verified using seven real datasets. Experimental results show that PLAS algorithm is superior to other link prediction algorithms.The existing algorithm PLAS is improved by introducing a graph attention network, and a link prediction algorithm (PLGAT) is proposed, which was verified using seven real datasets and two 5G/6G space–air–ground communication networks. The experimental results show that the PLGAT algorithm is superior to other link prediction algorithms. Furthermore, our proposed PLGAT algorithm for link prediction can precisely find out the new links on the Mobile MEC equipment network in 5G/6G to provide better QoS for data transportation.

## 3. PLAS Model Framework

This paper proposed a link prediction algorithm PLAS (predicting links by analysis subgraph) based on subgraphs. The PLAS algorithm transforms the link prediction task into a graph classification task, with the target node taking the linked subgraph as a positive sample and the target node taking the unlinked subgraph as a negative sample. Compared with the link prediction algorithm based on machine learning, it uses node labels to learn the graph features of the subgraph automatically and integrates the potential features of the subgraph nodes and node attribute features. The subgraph with multiple features has more comprehensive information, which is able to improve link prediction accuracy. Figure 1 is the frame diagram of the model, which is mainly divided into four modules: 1. extraction of subgraphs; 2. graph labeling algorithm; 3. subgraph encoding; and 4. fully connected layer learning.

The principle of the PLAS algorithm is shown in Figure 1. Given the undirected and unweighted graph $G\left(V,E\right)$ at the input time t, the subgraph of the h-hop of the target node A, B is first extracted, namely $G_{A,B}^{h}$ (1-hop in the graph), and then labels are assigned to each node in the subgraph through the graph-labeling algorithm. The nodes in the subgraph are sorted according to labels and the information characteristic matrix $X_{A,B}^{h}$ is constructed according to their order. Then, the first k node is selected, splicing the corresponding features of the first k node in the information matrix $X_{A,B}^{h}$ to represent the embedding of the subgraph of target node pair, and finally input it to the fully connection layer for learning. $G_{x,y}^{h}$ is the closed subgraph of the h-hop of the target node pair (x,y), including all the first-order, second-order, and h-order nodes and the edges of corresponding nodes. $X_{x,y}^{h}$ is an information feature matrix that sorts nodes in the subgraph and builds in accordance with their order, which is described as follows:(1)$$X_{A,B}^{h}=\left(\begin{array}{ccc} F_{1}^{e} & F_{1}^{l} & F_{1}^{g}\\ \vdots & \vdots & \vdots \\ F_{n}^{e} & F_{n}^{l} & F_{n}^{g} \end{array}\right)$$
where $F_{1}^{e}$, $F_{1}^{l}$, and $F_{1}^{g}$ are the explicit attribute feature, implicit attribute feature, and graph label of the first node in the information feature matrix. Additionally, each row in information feature matrix $X_{x,y}^{h}$ represents the feature of a node in the subgraph $G_{x,y}^{h}$. The feature of each node is composed of explicit attributes (attributes of the node itself, such as interest, position, gender, etc.), implicit attributes (Node2Vec [31] embedding nodes in the graph) and the label of each node in the subgraph (network structure features).

### 3.1. Extraction of Subgraphs

In order to learn the topological features of the network, the PLAS algorithm extracts its h-hop closed subgraph for each target node pair. In a natural system, the connection between nodes is affected by neighbor nodes, and the more intermediate nodes on the node path, the weaker the relationship between two nodes. Although more node structure may lead to more link information, whether the node pair will generate a link is primarily affected by 1-hop or 2-hop neighbor nodes. In this paper, the h parameter is set to 1 or 2, according to the sparseness of the network structure. When the average number of neighbor nodes of each node in the network is greater than 10, parameter h is selected as 1; when the average number of neighbor nodes of each node in the network is less than 10, parameter h is selected as 2. This ensures that the value of h will not be set too small, resulting in insufficient neighbor information and thus significantly impacting the prediction results. Moreover, with the decrease of the h parameter, the number of neighbor nodes extracted from the target node pairs will decrease exponentially, indicating that the dimension of the feature vector of subgraph decreases, which can significantly reduce the training time of the model.

However, for large-scale networks, such as social networks, where a superstar has millions of fans, the number of first-order neighbor nodes is very large, which will lead to a memory explosion when the subgraph is extracted. This paper sets a threshold N to prevent the problem of oversampling of subgraphs. When extracting the subgraph of the target node pair, if the number of subgraph nodes exceeds the threshold value N, the current h-order neighbor nodes will be randomly sampled to equal N, prioritizing sampling nodes with closer hops. For example, we need to extract 2-hop neighbor node subgraphs, while setting the threshold N value with 100, extract first-order neighbor nodes to 50 and extract second-order neighbor nodes to 250, and the sum of first-order neighbor nodes and second-order neighbor nodes is thus greater than the threshold N. Therefore, we need to randomly sample second-order neighbor nodes (randomly select 50 nodes from 250 nodes) so that the number of summary points in the subgraph is equal to the threshold N.

The subgraph extraction algorithm is described as Algorithm 1:
**Algorithm 1:** Subgraph extraction**Input:**  Target node pair i,j, graph GV,E, h-hops neighbor nodes, threshold N**Output:** h-hops subgraph S of target node pair i,j1: V(s)={i,j}2: N(temp)={}3: c=14: while c≤h do5:    N(temp)=Tic∪Tjc6:    if len(N(temp))+len(V(s))>N:7:         N(temp)=sample(N(temp))8:         V(s)=N(temp)∪V(s)9:         break10:   V(s)=N(temp)∪V(s)11:   c=c+112: end while13: S=SubGraph(V(s))14: return S

The target node h-hop neighbor node subgraph extraction algorithm process is as follows: for the target node pair $\left(i,j\right)$ in graph G, first add the target node pair (i,j) to the subgraph node set V(s), and then add the node pair (i,j), the first-order neighbors, and the second-order neighbors to the h-order neighbors to the set V(s). The relationship between all nodes and nodes in the set constitutes a subgraph S.

### 3.2. Graph Labeling Algorithm

The graph is an unordered data structure. We need a graph labeling algorithm to sort the nodes in the graph according to their labels to form an ordered sequence so that the fully connected layer can read node feature in a consistent order.

The Weisfeiler–Lehman (WL) algorithm [43] is widely used in graph classification, and it is a classic graph labeling algorithm. The main idea of WL algorithm is to use the labels of neighbor nodes to update their own labels iteratively and compress the updated labels into new ones until they converge. The primary process of the WL algorithm is as follows:It initializes all nodes in the graph to the same label 1, and each node aggregates its label and the labels of neighbor nodes to construct a label string.The nodes in the graph are sorted in ascending order of label strings, and according to the sorting update to new labels 1, 2, 3, ... nodes with the same label string will receive the same new label. For example, suppose that the label of node x is 2, its neighbor label is {3,1,2}, the label of node y is 2, and its neighbor label is {2,1,2}. The label strings of x and y are < 2123 > and < 2122 >, respectively. Because < 2122 > is less than < 2123 > in the dictionary order, y will be assigned a smaller label than x in the next iteration.This process is repeated until the node label stops changing. Figure 2 shows updating the nodes’ labels from 1 to (1, ... 5).

The WL algorithm has two key advantages: 1. The final label coding represents the structural role of nodes in the graph. Nodes with similar structures have similar labels in different graphs. 2. It defines the relative order of nodes in the graph, which is also consistent in different subgraphs. However, the WL algorithm treats all nodes in the subgraph equally during initialization and assigns the same label. After multiple rounds of iterations, the final node label encoding makes it impossible to distinguish the target node pair from other nodes in the subgraph, resulting in untargeted model training. Therefore, we designed a new graph labeling algorithm combined with the shortest distance, which can ensure that the nodes in different subgraphs are sorted in a consistent order and distinguish the target node pair from other nodes in the subgraph.
**Algorithm 2:** subgraph labeling**Input:** Target node pair i,j, subgraph node list N, subgraph S **Output:** Ordered list with labels O1:  O←(i,(0,1))2:  O←(j,(1,0))3:  R=N−{i,j}4:  for each v∈R do5:     di=d(v,i)6:     dj=d(v,j)7:     O←(v,(di,dj))8:  end for9:  sorted O by (di,dj)10: return O

The process of the subgraph labeling algorithm is as Algorithm 2. The label string $\left(d_{i},d_{j}\right)$ for each node consists of the shortest distance from that node to the target link node pair (i,j), where $d_{i}$ and $d_{j}$ are the shortest distances from this node to the target nodes i and j, respectively. After extracting the subgraph, we put all the extracted nodes into set R. For each node in set R, we calculate the shortest distance $\left(d_{i},d_{j}\right)$ from each node to the target node pair (i,j) as the label string and add it to the sequence list O for uniform sorting. Additionally, we allocate the label strings $\left(0,1\right)$ and $\left(1,0\right)$ for the target link node pair (i,j), respectively. The sequence list O is sorted first using the value of $d_{i}$ and then using the value of $d_{j}$.

Since the shortest distance $\left(d_{i},d_{j}\right)$ from other nodes to the target node pair is always greater than or equal to 1, according to the sorting rules of the sequence list, the label string (0,1), $\left(1,0\right)$ of the target link node pair is always smaller than the label string of other nodes. Therefore, the first two elements in the sequence list O are always target node pairs (i,j), which can well distinguish the target link node from other nodes in the subgraph. The label string of a node defines the structural role in the subgraph centered on the target link node pair. Additionally, in the subgraph centered on different links, nodes with the same structural role have similar label strings. Compared with the WL algorithm, the graph labeling algorithm in this paper only needs to calculate the shortest distance from the node in the subgraph to the target node, and there is no iterative process of updating the node label of the sub graph, so it has low time complexity.

### 3.3. Subgraph Encoding

The purpose of subgraph encoding is to represent the subgraph centered on the target node pair as a node information feature matrix $X_{x,y}^{h}$ with a specific order. We construct the information feature matrix according to the order of nodes in the sequence list O. The information feature of each node is composed of an explicit attribute, implicit attribute, and node label.

Explicit attribute: The attribute of the node itself is the attribute feature of the dataset. For example, in the social network, the attribute feature of the node is interest, hobby, gender, etc.

Implicit attribute: Forming node embeddings in a graph using a graph embedding algorithm. Node2vec is a common node embedding algorithm. It first obtains a series of node sequences via random walk in the network, and then obtains models, such node sequences, by processing word vectors to obtain the representation of node vectors in the network.

The label of each node in the subgraph: The node label is represented by the function $f(v)=(d_{i},d_{j})$, which assigns a label string f(v) to each node in the subgraph. Node labels have three main functions:They can represent different nodes playing different roles in the subgraph. The shorter the shortest path of other nodes in the subgraph relative to the target node pair, the greater its impact regarding whether the target node will generate links in the future, so it plays a more important role in the subgraph.The graph is an unordered data structure, which has no fixed order. Therefore, it is necessary to sort the nodes in the subgraph through labels and then input them to the fully connection layer in a consistent order for learning.After extracting the h hop subgraph of the target link node pair (i,j), we calculate the shortest path from other nodes in the subgraph to the target node pair (i,j) and assign a label string to each node in the subgraph. When all node features in the subgraph are spliced and sent to the fully connected layer for learning, the fully connected layer will automatically learn the graph structure features suitable for the current network, including the discovered graph structure features or the undiscovered graph structure features. For example, the CN algorithm calculates the number of common neighbor nodes of the target node pair. The full connection layer only needs to find the number of nodes with node label (1,1). By assigning a node label string to each node in the graph through the icon algorithm, our algorithm model can automatically learn the graph structure characteristics of the network, so it can be applied to different network structures. The later experimental results show that our algorithm is better in the AUC (area under curve) than other link prediction algorithms.

When we obtain the explicit attribute, implicit attribute, and label of each node in the subgraph, we splice the characteristics (explicit attribute, implicit attribute, and node label) of each node in the order of the sequence list and construct an information feature matrix to represent the whole subgraph.

### 3.4. Fully Connected Layer Learning

The fully connected layer is used to integrate the information feature matrix $X_{x,y}^{h}$ of the h hop subgraph of different target link node pairs for learning. Since the number of nodes in the h-hop subgraph of each node pair is different, the training of the fully connected layer will be significantly affected by the subgraph with a large number of nodes, while the impact of the node information with a small number of subgraphs will be affected by neglect. Therefore, it is necessary to set a unified node number threshold K to balance the feature dimension input of different subgraphs. We sort each node in the subgraph using a graph-labeling algorithm. When the number of nodes in the subgraph is greater than K, the nodes ranked after K are discarded. When the number of nodes in the subgraph is less than K, we construct virtual nodes so that the number of nodes equals K, and zero vectors represent the feature of the virtual nodes in $X_{x,y}^{h}$. We set the specific K value according to the overall network topology. We sort the number of each subgraph node in a training set in ascending order and take the value of the corresponding element at the 0.6 scale index as K. If the K value is set too small, too many nodes need to be deleted, resulting in too much information loss; if the K value is set too large, we need to construct a large number of virtual nodes that do not contain any information, and this will also cause the input feature dimension to be too large, increasing the training time. Therefore, we have compromised selecting the K value according to the network topology.

The neural network structure of the fully connected layer consists of an input layer, a hidden layer, and an output layer. The number of neurons in the input layer is determined by the dimension of each node feature vector multiplied by K, and the number of neurons in the hidden layer is set to 128. Additionally, the number of neurons in the output layer is 2, because we converted link prediction into a graph classification problem (two classes), where the subgraph with linked node pairs represents one class and the subgraph without linked node pairs represents the other class. In this paper, the ReLU function was used for activation function and cross entropy loss function for the loss function.

## 4. Experimental Results

Experimental environment: A stand-alone processor AMD Ryzen 7 4800H with Radeon Graphics 2.90 GHz, NVIDIA GeForce RTX 2060, and a memory of 16G was used. The operating system was Windows 10, the programming language is Python, and the system was based on the deep learning framework Pytorch.

### 4.1. PLAS Algorithm

We conducted experiments with the PLAS model on seven real datasets and evaluated our model with AUC (area under curve). USAir [44] is the transportation network of American Airlines, NS is the co-author network composed of researchers on the network [45], Pb is the blog network of the U.S. government [46], Yeast is the network composed of protein interactions in yeast [47], Cele is the divine network composed of neurons of a worm [48], Power is the power network in the western United States [48], and Router is the Internet network composed of routers [49]. Furthermore, we introduced the dataset Bupt5GMEC, which is the real network structure of the 5G MEC access network at BUPT (Beijing University of Posts and Telecommunications) and demonstrates the inside routing structure of BUPT for 5G MEC experimental environment access through a MEC Hub Node. Additionally, this 5G MEC experimental environment access was used to evaluate routing links in 5G MEC experiments in Figure 3. Aside from this, we also found = other important 5G MEC access networks with named CM1-5G and CM2-5G from other 5G MEC experiments, jointly developed with China Mobile. Finally, a real AS topology global internet dataset ITDK0304S was added to Table 1 to carry out an evaluation on the largest subset of real internet topology [50]. For the construction of the ITDK0304S dataset, because the original dataset has some nodes separated from the overall network structure, there are few links between these nodes, and they are relatively independent of the overall network. As a result, it contained less information and has lower research value, which affected model training. Therefore, we used the FastGN [51] community detection algorithm to divide the original dataset. We used several communities that mainly represented the overall structure to construct the ITDK0304S dataset and delete certain individual communities with few nodes and nodes not worth researching. Since our model performs link prediction through extracted subgraphs, for large-scale real-world networks, we were able to adjust the hop count and the threshold N of sampling neighbors during subgraph extraction to ensure sufficient information for training without causing oversampling.

Table 1 contains the statistical information of each network. We abstracted the real-world network structure data into the form of point-edge pairs. We took all the node pairs with edges in each network (as positive samples) and took the same number of node pairs without edges (as negative samples) to form a dataset, and then randomly selected 90% of the node pairs with and without edges as the training set and the remaining 10% of the dataset as the test set. When we used the Node2Vec algorithm to calculate the implicit attributes of nodes, as the node sequence obtained by random walk may imply information about whether there is a link between the target node pair, resulting in poor model generalization. Therefore, when the Node2Vec algorithm embedded nodes for each subgraph, links were added to the target nodes without a link relationship so that the link information of any target node pair in the training set was the same.

We selected five classical heuristic algorithms based on node similarity to compare with our algorithm, namely common neighbors (*CN*) [45], Jaccard coefficient (*JC*) [13], Adamic ADAR (*AA*), resource allocation (*RA*), and *Katz* [52].

*CN*: the number of common neighbors of target node x and y, which is defined as follows:(2)CN(x,y)=|T(x)∩T(y)|

T(x) represents a collection of neighbor nodes x, |T(x)| represents the number of neighbors of node x.

*JC*: similar to the *CN* algorithm. It is the expression of standardization of common neighbors. Its definition is as follows:(3)$$JC(x,y)=|\frac{T(x)\cap T(y)}{T(x)\cup T(y)}|$$

*AA*: a neighbor node with a small degree has a larger weight, which is defined as follows:(4)$$AA(x,y)=\sum_{z\in T(x)\cap T(y)}\frac{1}{\log|T(z)|}$$

*RA*: similar to *AA*, but the punishment is different. Its definition is as follows:(5)$$RA(x,y)=\sum_{z\in T(x)\cap T(y)}\frac{1}{\left|T(z)\right|}$$

*Katz*: calculates all paths of the target node. The shorter the path, the higher the weight. It is defined as follows:(6)$$Katz(x,y)=\sum_{l=1}^{\infty }B^{l}\cdot |path_{x,y}^{l}|$$

We also selected three methods of link prediction using graph embedding to compare with our model, which are node2vec (n2v) [31], line [30], and vgae [35]. These all embed the nodes of the graph, then splice the vectors of the target node pairs and send them to the fully connected layer for classification learning.

We use 50 epochs to train our model. In the process of training, we randomly select 10% of the training set as our verification set for evaluation, and we save the model with the best performance on the verification set. Finally, we use the saved model to predict the test set.

We first compared the five heuristic-based link prediction algorithms, and the results are shown in Table 2 and Figure 4:

As shown in Table 2, our algorithm performs better than heuristic link prediction algorithm on the datasets NS, Pb, Yeast, Cele, Power, and Router. On the USAir dataset, our algorithm also performs better than Katz and JC algorithms. Of these, on sparse datasets such as Power and Router, the AUC of heuristic link prediction algorithm is around 65% and 55%, respectively, while the AUC of our link prediction algorithm reaches around 75% and 85%, which an increase of 10% and 30%, respectively

We then compared three link prediction algorithms based on graph representation. The experimental results are shown in Table 3 and Figure 5:

As shown in Table 3, the link prediction algorithm based on graph representation has better performance on the sparse dataset of power and router than the heuristic link prediction algorithm, but compared with our algorithm, especially on the router data, our algorithm improves the AUC by about 20%. On USAir, NS, Pb, Yeast and Cele datasets, our algorithm is better than the link prediction algorithm based on graph representation.

### 4.2. PLGAT Algorithm

The PLAS algorithm treats neighbor nodes equally, lacks the consideration of the degree of influence on different neighbor nodes, and may discard some nodes in the training of the fully connected layer, resulting in information loss. Aiming at these problems, we improve PLAS and propose a link prediction algorithm PLGAT (predicting links by graph attention networks) based on graph attention network [36]. The algorithm distinguishes the degree of influence of different nodes on the target node through the attention mechanism. It aggregates the information of the discarded nodes through two layers of GAT convolution layers so that the nodes have comprehensive information and improve the accuracy of the link prediction algorithm. In addition, the algorithm defines the relative position of the nodes in the subgraph through the pooling layer module and reflects the directionality of the subgraph (centering on the target node and expanding to the neighbor nodes on both sides), which improves the link prediction results. The PLGAT model framework of our algorithm is shown in Figure 6:

Compared with the PLAS model, PLGAT uses the subgraph features for further processing with GAT. First, the h-hop subgraph of the target node pair was extracted. Then, the subgraph aggregated the information of the neighbor nodes through the GAT convolution layer and combined the original node features with the new features obtained through the GAT convolution layer to obtain the new feature representation of the node. Finally, through a pooling layer, nodes in the subgraph are sorted according to specific sorting rules and the features of the first K nodes are selected as the subgraph encoding, which is sent to the full connection layer for classification.

#### 4.2.1. GAT Convolution Layer

Graph attention network (GAT) takes a set of node features $h=\{{\vec{h}}_{1},{\vec{h}}_{2},\ldots ,{\vec{h}}_{N}\}$ as input, where ${\vec{h}}_{i}\in R^{F}$, N represents the number of nodes and F indicates the dimension of the node’s original features. It generates a new set of features $h^{\prime }=\{{\vec{h}}_{1}^{\prime },{\vec{h}}_{2}^{\prime },\ldots ,{\vec{h}}_{N}^{\prime }\}$ through the graph attention layer, where ${\vec{h}}_{i}^{\prime }\in R^{F^{\prime }}$, $F^{\prime }$ indicates the dimension of the node’s new feature vector.

In order to obtain sufficient expression information, GAT needs to perform at least one learnable linear transformation on the features of input nodes to obtain new node features. Therefore, the algorithm trains a weight matrix w for all nodes, $w\in R^{F^{\prime }\times F}$, this weight matrix is the relationship between input features and output features. For each node, a single-layer feed-forward neural network is used to calculate the self-attention coefficient between nodes. The attention coefficient is expressed as:(7)$$e_{ij}=a(W{\vec{h}}_{i},W{\vec{h}}_{j})$$

a is a single-layer feed forward neural network, and $e_{ij}$ represents the importance of node j to node i. Typically, each node computes the attention coefficient with any other node, but this results in a loss of graph structure information. In order to obtain the graph structure features, GAT introduces the masked attention mechanism and distributes attention to the set of neighbor nodes $N_{i}$ of node i and $j\in N_{i}$. At the same time, GAT uses the SoftMax function to regularize all neighbor nodes j of node i, and adds a LeakyReLU nonlinear function to the output of the feed forward neural network. Therefore, the formula of the complete attention mechanism is as follows:(8)$$a_{ij}=\frac{\exp(\mathrm{LeakyReLu}({\vec{a}}^{T}[W{\vec{h}}_{i}||W{\vec{h}}_{j}]))}{\sum k\in N_{i}\exp(\mathrm{LeakyReLu}({\vec{a}}^{T}[Wh_{i}||W{\vec{h}}_{k}]))}$$

a is a single-layer feedforward neural network, and $\vec{a}$ is the weight matrix between layers in the neural network.

After obtaining the attention coefficient between different nodes through the above formula, the final output characteristics of the node are obtained by aggregating the characteristics of neighboring nodes. The formula is as follows:(9)$${\vec{h}}_{i}^{\prime }=\sigma (\sum_{j\in N_{i}}a_{ij}W{\vec{h}}_{j})$$

σ is a non-linear activation function.

In order to stabilize the learning process of self-attention, GAT uses a multi-head attention mechanism. Specifically, it uses k independent self-attention mechanisms to splice the obtained new features. GAT uses k average instead of splicing operations. Therefore, the final formula of the multi-head attention mechanism is:(10)$${\vec{h}}_{i}^{\prime }=\sigma (\frac{1}{K}\sum_{k=1}^{K}\sum_{j\in N_{i}}a_{ij}^{k}W^{k}{\vec{h}}_{j})$$

k represents a total number of attention mechanisms, $a^{k}$ represents the kth attention coefficient, and $W^{k}$ represents the linear transformation weight matrix under the kth attention mechanism.

In the task of deep learning, it is generally believed that the higher the complexity of the model, the higher the fitting effect. For the graph neural network, whether to add more convolutional layers to improve the model’s performance should be decided according to the specific situation of the task. Increasing the depth of the convolutional layer increases the aggregation radius, but when the convolutional layers reach a certain depth, almost every node in the graph contains global graph information. Although a large number of convolutional layers can bring global information beneficial to graph classification tasks, it increases the complexity of the model, resulting in a long training time and model overfitting. Since the category of a node is primarily affected by local neighbor nodes, a small number of convolutional layers can achieve better performance, so we only extract the target node to the neighbor nodes within two hops to form a subgraph, and only use two convolutional layers to cause the target node to aggregate with the global graph information, which improves the accuracy of the algorithm without resulting in the complexity of the model.

We use the node features obtained in the subgraph encoding part of PLAS as the original input features $F^{0}$. The original feature $F^{0}$ is input into the GAT convolution layer for calculation, the feature $F^{1}$ is obtained through the first layer of GAT convolution, and the feature $F^{2}$ is obtained through the second layer of the GAT convolution. We concatenate the three feature to obtain the final feature representation of the node as follows:(11)$$F=F^{0}||F^{1}||F^{2}$$

Of these, || represents the connection operation, and the final feature representation of each node obtained through two GAT convolutional layers contains almost the global information of the subgraph.

#### 4.2.2. Pooling Layer

The pooling operation of the graph is used to aggregate the global node information to obtain a global graph vector representation. The pooling methods of graph data are mainly divided into two categories, one is hierarchical pooling [53] and the other is global pooling [54]. The hierarchical pooling method mainly includes two processes: 1. the assignment matrix is calculated to determine which neighbor nodes are allocated together to form a cluster and 2. the nodes in each cluster are aggregated into a new super node in a certain aggregation form, and then these super nodes are used to form a new graph. For large-scale graphs, this hierarchical pooling method is very effective. The scale of the graph is reduced through continuous convolution and pooling operations to obtain the final representation of the entire graph. The global pooling method selects appropriate nodes to form a representation of the entire graph after multiple convolutions, which minimizes the loss of information. Therefore, in s small-scale graph, the global pooling method is superior to the hierarchical pooling method. We extract the h-hop neighbor nodes of the target node pair to form the subgraph, which mainly has the following two characteristics:The subgraph is centered on the target node pair and spreads out to the neighbor nodes on both sides, which has strong directionality.The number of neighbor hops in the subgraph is limited, resulting in the subgraph often being a small-scale graph.

For small-scale graphs, using the global pooling method is more appropriate. However, it is difficult for this global pooling method to show the directionality of the subgraph, and it is challenging to select appropriate nodes with uniform rules to form a subgraph feature representation. Based on the global pooling method, we use the graph labeling algorithm in PLAS to sort the nodes in the sub-graphs by a specific sorting rule to solve the above problems. In graph labeling algorithm, the label string $\left(d_{i},d_{j}\right)$ consists of two parts, one is the shortest distance $d_{i}$ from the target node i and the other being the shortest distance $d_{j}$ from the target node j. For the target node i, the label character is always (0,1), and for the target node j, the label string is always (1,0). The nodes in the subgraph are first sorted by $d_{i}$, and then sorted by $d_{j}$ if $d_{i}$ is equal. Sorting using this rule has two main benefits:Because the shortest distance $d_{i}$ from other nodes in the subgraph to the target node pair (i,j), $d_{j}$ is greater than or equal to 1, the target node pair (i,j) is always ranked in the first two elements, and the node closer to the target node in the subgraph will be ranked in the front, and the node farther away from the target node will be ranked in the back, which reflects the direction of the subgraph.Nodes in the subgraph will be sorted according to consistent rules. We only need to select the first K nodes to represent the feature expression of the current subgraph, to unify the rules of node selection in the global pooling method. In the case that the number of nodes in the subgraph is greater than K, only nodes after K need to be truncated, and the discarded nodes will not lose all node information, because other nodes in the subgraph have learned the discarded node information through the two layers of the GAT convolution layer. If the number of nodes in the subgraph is less than K, simply add virtual nodes, where virtual nodes are represented by the zero vector.

#### 4.2.3. Experimental Dataset and Settings

We carried out experiments on seven real datasets, namely USAir [44], NS [45], PB [46], Yeast [47], Cele [48], Power [48], and Router [49], and evaluated the performance of the algorithm with the AUC.

To evaluate our algorithm on real 5G MEC access network environment and real AS global internet topology, we tested our algorithm on Bupt5GMEC, CM1-5G, CM2-5G, and ITDK0304S network datasets.

For each dataset, all the linked node pairs in the current network structure were selected as positive samples, and then an equal number of unlinked node pairs were randomly selected as negative samples, which balances the positive and negative samples. Additionally, 80% of the positive samples and 10% of the negative samples were randomly selected as the training set, 10% as the verification set, and the remaining 10% as the test set. The number of neurons in the first hidden layer of the fully connected layer was set to 1024, the number of neurons in the second hidden layer was set to 512, and the number of neurons in the output layer was set to 2.

#### 4.2.4. Experiment Results Analysis

We use the AUC as evaluation of algorithm, during the training process, we used 50 epochs to train our model, saved the model that performs best on the verification set, and used it for prediction on the test set. We repeated the above operation ten times and took the average of the ten experimental results as the final result.

Firstly, we compared five heuristic-based link prediction algorithms, and the experimental results are shown in Table 4:

As shown in Table 4 and Figure 7 and Figure 8, our methods, PLAS and PLGAT, are better than the heuristic link prediction algorithm in the NS, PB, Yeast, Cele, Power, Router, Bupt5GMEC, CM1-5G, CM2-5G, and ITDK0304S datasets. Especially in the dataset with sparse data, such as the Power, Router, and Bupt5GMEC communication networks, our methods are obviously superior to the heuristic link prediction algorithm, while the AUC index increased by about 15% and 30%, respectively. On the USAir dataset, our method performs as well as the heuristic link prediction algorithm, which is slightly lower than the RA algorithm and better than other algorithms.

PLGAT was compared with three link prediction algorithms based on graph representation learning and the PLAS algorithms, and the results are shown in Table 5:

As shown in Table 5 and Figure 9, compared with PLAS, both algorithms convert link the prediction into graph classification tasks. The PLGAT algorithm proposed in this chapter is superior to PLAS algorithm in the USAir, PB, Cele, Power, Router, Bupt5GMEC, CM1-5G, CM2-5G, and ITDK0304S datasets. In the Power dataset, our algorithm improved by more than 5%. In terms of the Yeast data, the performance of our algorithm was inferior to that of the PLAS algorithm, but the difference was within 1%.

Compared with VGAE algorithm, both algorithms use GNN (graph neural network) to learn neighbor node information. Our algorithm is superior to the VGAE algorithm in USAir, NS, PB, Yeast, Cele, Power, Router, Bupt5GMEC, CM1-5G, CM2-5G, and ITDK0304S datasets. On the Router dataset, our algorithm improved by more than 30%. For the LINE and Node2Vec algorithms, our algorithm outperforms these in all respects. For the heuristic link prediction algorithm, the performance of our algorithm is lower than that of RA algorithm only in the USAir dataset, and in other datasets, the performance of our algorithm is far better than that of the heuristic link prediction algorithm.

## 5. Conclusions

Link prediction is a hot research field at present. It is very important for mining and analyzing the evolution of networks. Although the heuristic algorithm based on node similarity is simple and effective, it cannot be applied to all network structures. Finding effective heuristic indexes for different network structures requires a process of repeated experiments. Based on the above problems, we design a link prediction algorithm based on target node pair subgraph, which combines the characteristics of graph structure and node characteristics and can play a role in different network structures. Finally, we compare five heuristic link prediction algorithms based on node similarity and three link prediction algorithms based on graph embedding on eleven real datasets. The results show that the performance of our algorithm is much better than other link prediction algorithms.

## Figures and Tables

**Figure 1 sensors-23-04936-f001:**
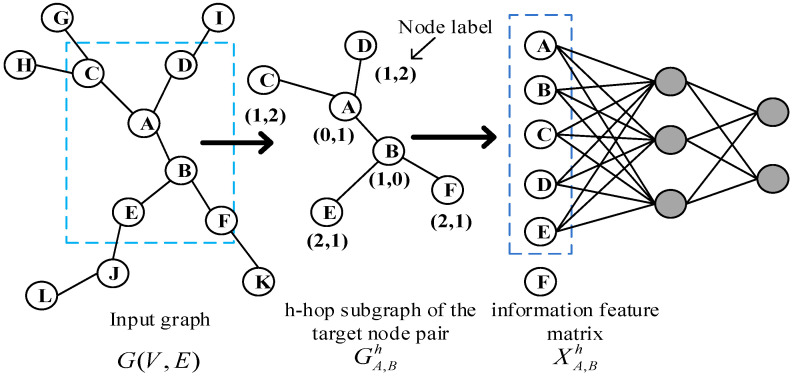
Schematic diagram of PLAS algorithm model.

**Figure 2 sensors-23-04936-f002:**
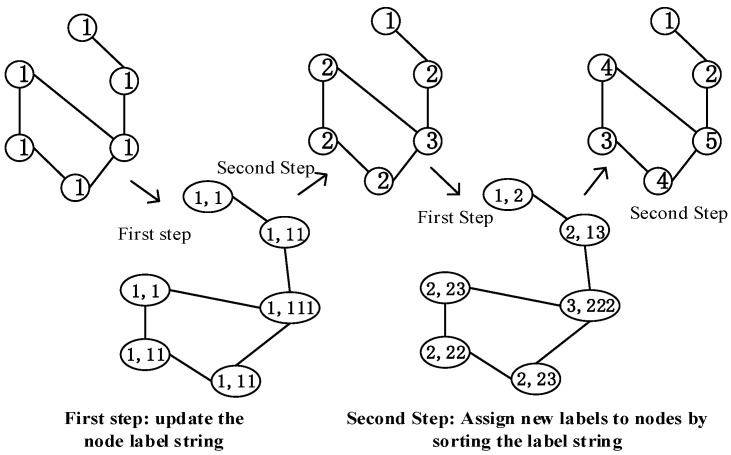
Schematic diagram of WL algorithm.

**Figure 3 sensors-23-04936-f003:**
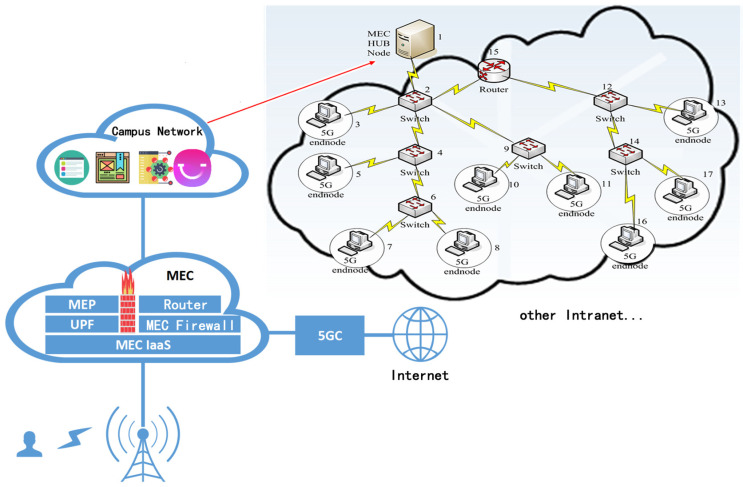
5G MEC access network structure in BUPT.

**Figure 4 sensors-23-04936-f004:**
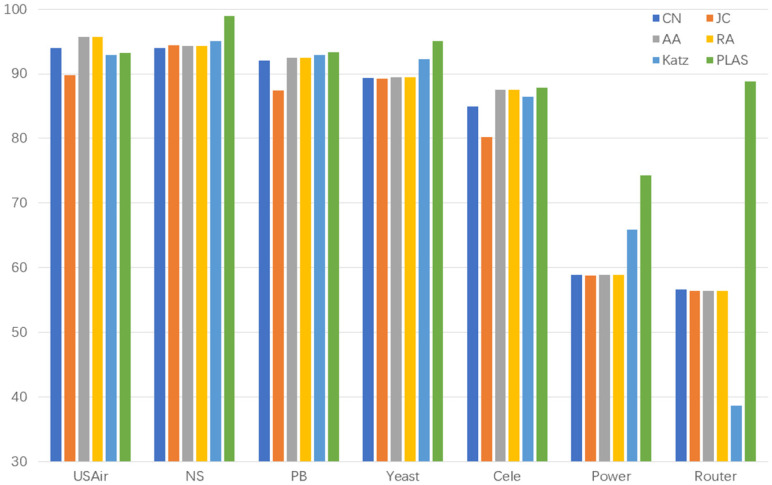
Comparison results with five heuristic link prediction algorithms (AUC).

**Figure 5 sensors-23-04936-f005:**
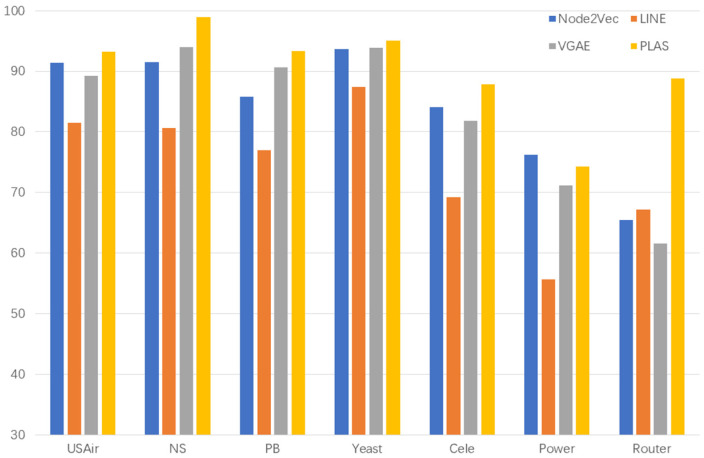
Comparison results with three link prediction algorithms based on graph representation learning (AUC).

**Figure 6 sensors-23-04936-f006:**
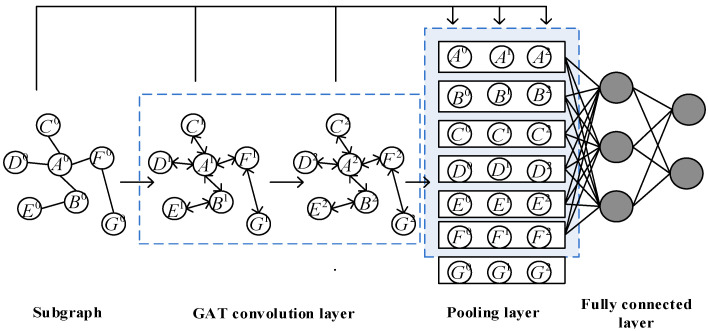
Schematic diagram of PLGAT algorithm.

**Figure 7 sensors-23-04936-f007:**
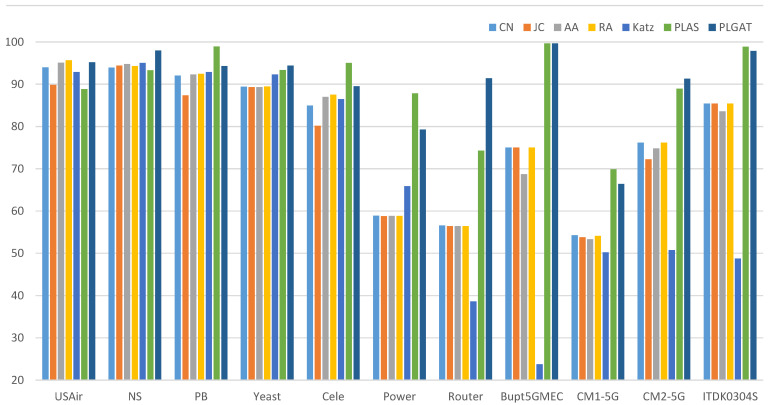
Comparison of PLGAT with five heuristic link prediction algorithms (AUC).

**Figure 8 sensors-23-04936-f008:**
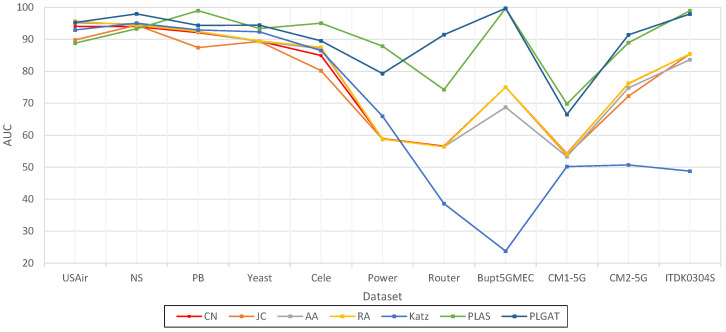
Comparison of PLGAT with five heuristic link prediction algorithms (AUC).

**Figure 9 sensors-23-04936-f009:**
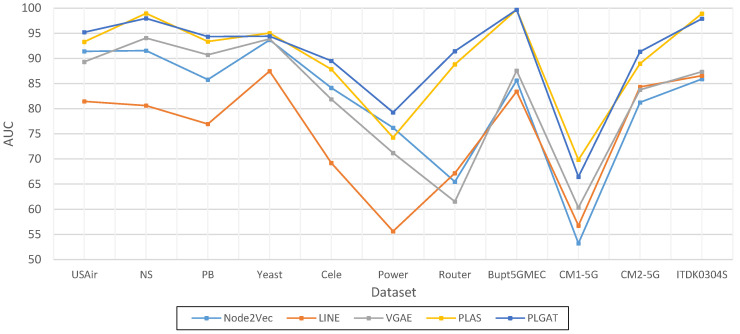
Graphic Comparison of PLGAT with three link prediction algorithms based on graph representation (AUC).

**Table 1 sensors-23-04936-t001:** Statistical information of network structure of experimental dataset.

Dataset	|V|	|E|
Router	5022	6258
USAir	332	2126
NS	1589	2742
PB	1222	16,714
Yeast	2375	11,693
Cele	297	2148
Power	4941	6594
Bupt5GMEC	135	338
CM1-5G	1500	5990
CM2-5G	1499	4498
ITDK0304S	3780	10,757

**Table 2 sensors-23-04936-t002:** Comparison results with five heuristic link prediction algorithms (AUC).

Dataset	CN	JC	AA	RA	Katz	PLAS
Router	56.57	56.38	56.40	56.38	38.62	88.81
USAir	94.02	89.81	95.08	95.67	92.90	93.28
NS	93.94	94.40	94.77	94.33	95.03	98.93
PB	92.07	87.39	92.31	92.45	92.89	93.37
Yeast	89.41	89.30	89.32	89.44	92.30	95.04
Cele	84.95	80.18	87.03	87.49	86.45	87.84
Power	58.90	58.80	58.83	58.83	65.90	74.27

**Table 3 sensors-23-04936-t003:** Comparison results with three link prediction algorithms based on graph representation learning (AUC).

Dataset	Node2Vec	LINE	VGAE	PLAS
Router	65.46	67.17	61.53	88.81
USAir	91.40	81.47	89.30	93.28
NS	91.55	80.63	94.04	98.93
PB	85.79	76.94	90.70	93.37
Yeast	93.68	87.45	93.87	95.04
Cele	84.13	69.22	81.87	87.84
Power	76.23	55.64	71.20	74.27

**Table 4 sensors-23-04936-t004:** Comparison of PLGAT with five heuristic link prediction algorithms (AUC).

Dataset	CN	JC	AA	RA	Katz	PLAS	PLGAT
USAir	94.02	89.81	95.08	95.67	92.90	88.81	95.21
NS	93.94	94.40	94.77	94.33	95.03	93.28	97.96
PB	92.07	87.39	92.31	92.45	92.89	98.93	94.32
Yeast	89.41	89.30	89.32	89.44	92.30	93.37	94.41
Cele	84.95	80.18	87.03	87.49	86.45	95.04	89.52
Power	58.90	58.80	58.83	58,83	65.90	87.84	79.27
Router	56.57	56.38	56.40	56.38	38.62	74.27	91.42
Bupt5GMEC	75.01	75.02	68.75	75.02	23.78	99.67	99.67
CM1-5G	54.23	53.79	53.32	54.13	50.22	69.84	66.43
CM2-5G	76.15	72.22	74.79	76.21	50.71	88.95	91.34
ITDK0304S	85.41	85.41	83.60	85.42	48.77	98.90	97.88

**Table 5 sensors-23-04936-t005:** Comparison of PLGAT with three link prediction algorithms based on graph representation (AUC).

Dataset	Node2Vec	LINE	VGAE	PLAS	PLGAT
USAir	91.40	81.47	89.30	93.28	95.21
NS	91.55	80.63	94.04	98.93	97.96
PB	85.79	76.94	90.70	93.37	94.32
Yeast	93.68	87.45	93.87	95.04	94.41
Cele	84.13	69.22	81.87	87.84	89.52
Power	76.23	55.64	71.20	74.27	79.27
Router	65.46	67.17	61.53	88.81	91.42
Bupt5GMEC	85.61	83.42	87.56	99.67	99.67
CM1-5G	53.22	56.75	60.34	69.84	66.43
CM2-5G	81.24	84.31	83.76	88.95	91.34
ITDK0304S	85.87	86.56	87.35	98.90	97.88

## Data Availability

The data presented in this study are available on request from the corresponding author. The data are not publicly available due to policy reasons.

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
