# Peer review of "Graph Neural Network-Based Efficient Subgraph Embedding Method for Link Prediction in Mobile Edge Computing"

_sensors, 2023, doi:10.3390/s23104936_

Round 1
Reviewer 1 Report
Link prediction is a hot research field at present. It is very important for mining and analyzing the evolution of networks. In order to mining and analyzing the evolution of network such for construction and analyzing logical architecture of MEC (Mobile Edge Computing) routing links of 5G/6G Access Network, this article has proposed a new efficient link prediction algorithm PLAS and its GNN (Graph Neural Network) version PLGAT based on subgraph of target node pair. This paper has tested its algorithm efficiency on several real 5G network structure and it still needs improvement:
1) The authors should revise its abstract part to more concentrated on the work this paper proposed and improve its grammar.
2) The authors must describe the PLAS and PLGAT models more detailed for reader to get the important point of model designing.
3) And the authors may supply more figure to demonstrate the comparison experiment results of different algorithms.
I recommend that authors use professional English proofreading services before publication.
Author Response
Response to Reviewer 1 Comments
Review 1 Comment:
Comments and Suggestions for Authors:Link prediction is a hot research field at present. It is very important for mining and analyzing the evolution of networks. In order to mining and analyzing the evolution of network such for construction and analyzing logical architecture of MEC (Mobile Edge Computing) routing links of 5G/6G Access Network, this article has proposed a new efficient link prediction algorithm PLAS and its GNN (Graph Neural Network) version PLGAT based on subgraph of target node pair. This paper has tested its algorithm efficiency on several real 5G network structure and it still needs improvement:
1) The authors should revise its abstract part to more concentrated on the work this paper proposed and improve its grammar.
2) The authors must describe the PLAS and PLGAT models more detailed for reader to get the important point of model designing.
3) And the authors may supply more figure to demonstrate the comparison experiment results of different algorithms.
Comments on the Quality of English Language:I recommend that authors use professional English proofreading services before publication.
Point 1: The authors should revise its abstract part to more concentrated on the work this paper proposed and improve its grammar.
Response 1: Thank you very much for pointing it out. In the revised manuscript, we have revise the abstract part with red marked words. And further more,we have checked the Englsih grammar in the whole paper and try our best to promote it.
Point 2: The authors must describe the PLAS and PLGAT models more detailed for reader to get the important point of model designing.
Response 2: Thank you very much for pointing it out. In the revised manuscript, we have revised and rewritten Section 3 and Section 4 marked by red words.
Point 3: And the authors may supply more figure to demonstrate the comparison experiment results of different algorithms.
Response 2: Thank you very much for pointing it out. In the revised manuscript, we have added some figures of Figure 8 And Figure 9.

Reviewer 2 Report
1st .The research contribution is not clear.
-The Research Objective is not clear.
2nd. The abstract should include Aim, Methodology and Results.
3rd. The abstract is unclear.
4th. This section is not clear:
5th.References: It is necessary to add a set of references because it is not enough so you need to add below References:
Hamad, A. A., Abdulridha, M. M., Kadhim, N. M., Pushparaj, S., Meenakshi, R., & Ibrahim, A. M. 2022. Learning methods of business intelligence and group related diagnostics on patient management by using artificial dynamic system. Journal of Nanomaterials, 2022, 1-8.
1st .The research contribution is not clear.
-The Research Objective is not clear.
2nd. The abstract should include Aim, Methodology and Results.
3rd. The abstract is unclear.
4th. This section is not clear:
5th.References: It is necessary to add a set of references because it is not enough so you need to add below References:
Hamad, A. A., Abdulridha, M. M., Kadhim, N. M., Pushparaj, S., Meenakshi, R., & Ibrahim, A. M. 2022. Learning methods of business intelligence and group related diagnostics on patient management by using artificial dynamic system. Journal of Nanomaterials, 2022, 1-8.
Reviewer 3 Report
This paper focus on link prediction problem in network science and tries to propose a method based on subgraph classification. The proposed method employs techniques from three aspects, include subgraph construction, subgraph encoding, fully connected layer. Experiments are conducted in representative datasets. The performace of the proposed methods are veirfied by comparing heuristic algorithms and representation learning based methods. Weaknesses: 1. The motivation, intuition, and the contributions are not represented clearly. The author spends a lot of space introduction the existing work and applications in Introduction Section, but does not clearly state what are the specific problems that the paper aims to solve for link prediction. 2. From line 153 to 164, "if the number of extracted subgraph nodes exceeds the threshold value ?, the current h-order neighbor nodes will be randomly sampled so that the number of subgraph nodes is equal to ?.". Do the seleted N nodes need to be connected, and how do you make sure they are connected? 3. "When the average number of neighbor nodes of each node in the network is greater than 10, parameter ℎ is selected as 1; when the average number of neighbor nodes of each node in the network is less than 10, parameter ℎ is selected as 2.". * The rationale behind this mechanism is not discussed in detail. 4. * The tenses of English sentences are inconsistent, for example: "We carried out experiments" "we compare with five heuristic based link" 5. The readability and the English writing of the study should be further improved, such as ", Nodes in the network represent entities" "Heuristic algorithm based on node similarity [7]." "Where ?1?, ?1? and ?1?"
Minor editing of English language required.The quality of English language meet the requirements of the journal, except for a few minor mistakes.
Author Response
Response to Reviewer 3 Comments
Review 1 Comment:
Comments and Suggestions for Authors:
This paper focus on link prediction problem in network science and tries to propose a method based on subgraph classification. The proposed method employs techniques from three aspects, include subgraph construction, subgraph encoding, fully connected layer. Experiments are conducted in representative datasets. The performace of the proposed methods are veirfied by comparing heuristic algorithms and representation learning based methods.
Weaknesses:
- The motivation, intuition, and the contributions are not represented clearly. The author spends a lot of space introduction the existing work and applications in Introduction Section, but does not clearly state what are the specific problems that the paper aims to solve for link prediction.
- From line 153 to 164, "if the number of extracted subgraph nodes exceeds the threshold value ?, the current h-order neighbor nodes will be randomly sampled so that the number of subgraph nodes is equal to ?.". Do the seleted N nodes need to be connected, and how do you make sure they are connected?
- "When the average number of neighbor nodes of each node in the network is greater than 10, parameter ℎ is selected as 1; when the average number of neighbor nodes of each node in the network is less than 10, parameter ℎ is selected as 2.". * The rationale behind this mechanism is not discussed in detail.
- * The tenses of English sentences are inconsistent, for example: "We carried out experiments" "we compare with five heuristic based link"
- The readability and the English writing of the study should be further improved, such as ", Nodes in the network represent entities" "Heuristic algorithm based on node similarity [7]." "Where ?1?, ?1?and ?1?"
Comments on the Quality of English Language:Minor editing of English language required.The quality of English language meet the requirements of the journal, except for a few minor mistakes.
Point 1: The motivation, intuition, and the contributions are not represented clearly. The author spends a lot of space introduction the existing work and applications in Introduction Section, but does not clearly state what are the specific problems that the paper aims to solve for link prediction.
Response 1: Thank you very much for pointing it out. In the Abstract part of revised manuscript, we have pointed it out as follows:
Link prediction can provide throughput guidance for MEC and select appropriate nodes for message forwarding processing and motivation of this article is construct better link prediction algorithm to optimize network structure and promote throughput MEC routing links of 5G/6G Access Network.
Traditional link prediction algorithms are always basing on node similarity which needing predefined similarity functions, is highly hypothetical and only can be applied to specific network structures without generality. To solve this problem, this paper has proposed a new efficient link prediction algorithm PLAS and its GNN (Graph Neural Network) version PLGAT based on subgraph of target node pair.
Point 2: From line 153 to 164, "if the number of extracted subgraph nodes exceeds the threshold value ?, the current h-order neighbor nodes will be randomly sampled so that the number of subgraph nodes is equal to ?.". Do the seleted N nodes need to be connected, and how do you make sure they are connected?
Response 2: Thank you very much for pointing it out. In the revised manuscript,we have revise it as this in Section 3.1:
When extracting the subgraph of the target node pair, if the number of subgraph nodes exceeds the threshold value , the current h-order neighbor nodes will be randomly sampled to equal , prioritizing sampling nodes with closer hops. For example, we need to extract 2-hop neighbor node subgraphs, while setting the threshold N value with 100, extract first-order neighbor nodes to 50, extract second-order neighbor nodes to 250, and the sum of first-order neighbor nodes and second-order neighbor nodes is greater than the threshold N. Therefore we need to randomly sample second-order neighbor nodes (randomly select 50 nodes from 250 nodes) so that the number of summary points in the subgraph is equal to the threshold N.
Point 3: "When the average number of neighbor nodes of each node in the network is greater than 10, parameter ℎ is selected as 1; when the average number of neighbor nodes of each node in the network is less than 10, parameter ℎ is selected as 2.". * The rationale behind this mechanism is not discussed in detail.
Response 3: Thank you very much for pointing it out. In the revised manuscript, we have modified to this:
When the average number of neighbor nodes of each node in the network is greater than 10, parameter is selected as 1; when the average number of neighbor nodes of each node in the network is less than 10, parameter is selected as 2. It ensures that the value of will not be set too small, resulting in insufficient neighbor information, significantly impacting the prediction results. Moreover, with the decrease of parameter, the number of neighbor nodes extracted from the target node pairs will decrease exponentially, indicating that the dimension of feature vector of subgraph decreases, which can significantly reduce the training time of the model.
The rationale behind this mechanism is that when average number of neighbor nodes of each node exceed 10, to avoid generating too much nodes in 2-hop subgraph, hop value h is limited to 1. On the contrary, hop value h is limited to 2.
Point 4: The tenses of English sentences are inconsistent, for example: "We carried out experiments" "we compare with five heuristic based link"
Response 4: Thank you for pointing it out. In the revised manuscript, we have followed your suggestion to check the Englsih grammar in the whole paper and try our best to promote it.
Point 5: The readability and the English writing of the study should be further improved, such as ", Nodes in the network represent entities" "Heuristic algorithm based on node similarity [7]." "Where ?1?, ?1? and ?1?"
Response 5: Thank you for pointing it out. Regarding the issues with the formulas, we have revised them according to your suggestions. As for the issue with abbreviations, we did not use a list of abbreviations but instead spelled out the full term when it first appeared in the manuscript and then included the abbreviation in parentheses. Further more , we rewrite the formula and mathmatics symbol in this paper all.
In link prediction, Nodes in the network always represent real entities such as routers and switches in network.
Heuristic link prediction algorithms are always based on node similarity.
Comments on the Quality of English Language:Minor editing of English language required.The quality of English language meet the requirements of the journal, except for a few minor mistakes.
Response : Thank you for pointing it out and we have checked the Englsih grammar in the whole paper and try our best to promote it.
Reviewer 4 Report
The following aspects should be addressed before further processing of this paper.
1. The reported solution is not properly compared with similar existing contributions, which exist. Thus, the authors should add a separate section, or a sub-section in "2. Related work", which would comparatively analyze their reported solution considering the advantages and drawbacks. For this purpose, at least three-four relevant existing papers should be selected, which reported similar significant contributions.
2. The list of references is rather undersized, especially considering the significance of the approached topic, and the existing papers. Therefore, another (at least) 10-15 additional references should be added, and properly analyzed in section 2, which must be extended accordingly. Additionally, there are papers that report interesting solutions, which can be applied to the study of complex networks of any kind, such as the one that is reported in https://univagora.ro/jour/index.php/ijccc/article/view/2198 . The authors may consider analyzing such papers.
3. The considered reference data sets are adequate, but the authors create small subsets out of the original data sets, or at least this is what their presentation suggests. Given the relatively small number of included vertices and edges(network links), the authors should discuss on their proposed model suitability to scale up well for large real-world settings.
4. The authors should also present the algorithm/methodology that was used in order to generate the experimental data sets.
5. The hardware and software specifications of the experimental infrastructure and not clear, they should be fully presented.
6. The English language should be fully proofread and improved, eventually using the help of a native English speaker.
Please see my comments.
Author Response
Response to Reviewer 4 Comments
Review 1 Comment:
Comments and Suggestions for Authors:
The following aspects should be addressed before further processing of this paper.
- The reported solution is not properly compared with similar existing contributions, which exist. Thus, the authors should add a separate section, or a sub-section in "2. Related work", which would comparatively analyze their reported solution considering the advantages and drawbacks. For this purpose, at least three-four relevant existing papers should be selected, which reported similar significant contributions.
- The list of references is rather undersized, especially considering the significance of the approached topic, and the existing papers. Therefore, another (at least) 10-15 additional references should be added, and properly analyzed in section 2, which must be extended accordingly. Additionally, there are papers that report interesting solutions, which can be applied to the study of complex networks of any kind, such as the one that is reported in https://univagora.ro/jour/index.php/ijccc/article/view/2198 . The authors may consider analyzing such papers.
- The considered reference data sets are adequate, but the authors create small subsets out of the original data sets, or at least this is what their presentation suggests. Given the relatively small number of included vertices and edges(network links), the authors should discuss on their proposed model suitability to scale up well for large real-world settings.
- The authors should also present the algorithm/methodology that was used in order to generate the experimental data sets.
- The hardware and software specifications of the experimental infrastructure and not clear, they should be fully presented.
- The English language should be fully proofread and improved, eventually using the help of a native English speaker.
Comments on the Quality of English Language:
Please see my comments.
Point 1: The reported solution is not properly compared with similar existing contributions, which exist. Thus, the authors should add a separate section, or a sub-section in "2. Related work", which would comparatively analyze their reported solution considering the advantages and drawbacks. For this purpose, at least three-four relevant existing papers should be selected, which reported similar significant contributions.
Response 1: Thank you very much for pointing it out. In the revised manuscript, we have revise the abstract part with red marked words. And further more,we have checked the Englsih grammar in the whole paper and try our best to promote it.
Point 2: The list of references is rather undersized, especially considering the significance of the approached topic, and the existing papers. Therefore, another (at least) 10-15 additional references should be added, and properly analyzed in section 2, which must be extended accordingly. Additionally, there are papers that report interesting solutions, which can be applied to the study of complex networks of any kind, such as the one that is reported in https://univagora.ro/jour/index.php/ijccc/article/view/2198 . The authors may consider analyzing such papers.
Response 2: Thank you very much for pointing it out. In the revised manuscript, We add some related work on link prediction based on machine learning and graph representation algorithms in the second part of the paper, which as followed:
“The link prediction algorithm based on machine learning mainly transforms the link prediction task into a binary classification task, in which the node pairs with links are re-garded as positive classes, and the node pairs without links are regarded as negative classes. The key to this kind of algorithm mainly lies in the selection of features and clas-sification algorithms. In 2004, Faloutsos et al. [32] introduced a connection subgraph, which can well capture the topology between two nodes in a social network. In 2006, Al et al. [33] extracted the non-topological features of the network based on extracting the topo-logical features of the network, which improved the algorithm's accuracy. In 2007, Liben et al. [34] extracted some network topology features from the citation network, such as CN, AA, Katz, etc., and input them into a supervised learning algorithm for learning and pre-diction. Their experimental results outperform link prediction algorithms based on indi-vidual network topologies. In 2010, Benchettara et al. [35] used the enhanced decision tree algorithm and found that using topological features in the feature set can significantly improve the link prediction algorithm's precision, recall, and F value. In 2014, Fire et al. [36] proposed a set of easily computable graph-structured features and adopted two en-semble learning methods to predict missing links in the network. In 2018, Zhang et al. [37] used the attribute features of nodes as non-topological features to input into supervised learning algorithms, improving link prediction algorithms' accuracy. In 2018, Mandal et al. [38] used a variety of supervised learning algorithms for link prediction in a two-layer network composed of Twitter and Foursquare. In 2022, Kumar et al. [39] introduced the value of node centrality as a sample feature, input it into various supervised learning al-gorithms for prediction, and achieved the best results on the LGBM (Light Gradient Boosted Machine) classifier. The key to link prediction algorithms based on machine learning lies in selecting feature sets. Such algorithms often extract some topological fea-tures of the network as feature sets. However, when solving domain-specific link predic-tion problems, corresponding domain knowledge is also required to construct its do-main-specific features. Compared with the heuristic-based link prediction algorithm, the link prediction algorithm based on machine learning can improve its accuracy, but it also brings time costs for training models and feature selection.
Link prediction algorithms based on graph representation learning mainly map high-dimensional dense matrices (graph data) into low-dimensional dense vectors and then use the mapped vectors for downstream tasks, such as node classification [40,41], graph classification [42,43], link prediction [44], etc. In 2014, Perozzi et al. [45] first pro-posed the graph representation algorithm DeepWalk for unsupervised learning. The algo-rithm obtains the sequence of nodes through random walks and inputs the sequence of nodes as sentences into the Skip-Gram model in the Word2Vec algorithm to obtain the node vector representation. In 2015, Tang et al. [23] proposed the LINE algorithm model, which proposed a method of edge sampling so that the vectors of the obtained nodes re-tain the first-order similarity and second-order similarity. In 2016, Grover et al. [13] pro-posed the Node2Vec algorithm. Node2Vec is similar to DeepWalk, but Node2Vec uses a biased random walk method, which balances depth-first walk and breadth-first walk, and obtains a higher-quality embedded representation. In 2016, Wang et al. [46] proposed the SDNE (Structural Deep Network Embedding) model. SDNE is a semi-supervised deep learning model that uses a deep network structure to simultaneously optimize the first-order and second-order similarity objective functions and obtains vectors for pre-serving the graph's local and global structure. In 2016, Cao et al. [47] proposed the DNGR algorithm model. DNGR uses the random walk model (Random Surfing) to generate the probability co-occurrence matrix, calculates the PPMI matrix with the probability co-occurrence matrix, and uses the superimposed denoising automatic encoding machine to extract features to obtain the vector representation of the node. In 2016, Kipf et al. [48] proposed the GCN (Graph Convolutional Network) model. The algorithm considers the influence of neighbor nodes and continuously aggregates the characteristics of neighbor nodes. Embedding neighbor nodes can obtain scalability, and the global information can be described by aggregating the characteristics of neighbor nodes through multiple itera-tions. In 2016, Kipf et al. [24] proposed the VGAE (Variational Graph Auto-Encoders) model, introducing variational autoencoders into graph data. The distribution of the node vector representation of the known graph is learned through GCN convolution, the repre-sentation of the node vector is sampled in the distribution and then decoded (link predic-tion) to reconstruct the graph. In 2017, Veličković et al. [28] proposed the GAT (Graph At-tention Networks) model, which introduced an attention mechanism. When calculating the vector representation of nodes, the model's generalization ability is improved by as-signing different weights to the characteristics of nodes. At the same time, a multi-head attention mechanism is introduced, and the features obtained by multiple attention mechanisms are spliced and averaged to obtain the final node representation. In 2017, Hamilton et al. [49] proposed the GraphSage algorithm model for large-scale graph data. By learning an aggregation function, the neighbor nodes are sampled and aggregated to obtain a new vector representation of the node. In 2018, Chen et al. [50] proposed the HARP algorithm model. HARP selects the starting node by weight and combines it with DeepWalk and Node2Vec to obtain a better embedding representation. In 2018, Schlicht-krull et al. [51] proposed the R-GCN (Relational Graph Convolutional Networks) algo-rithm model. R-GCN introduced weight sharing and parameter constraints to improve the performance of the link prediction algorithm. In 2018, Sam et al. [52] learned the repre-sentation vectors of nodes in historical time slices through the Node2Vec algorithm, con-catenated the vectors of nodes in historical time slices to obtain the representation of future time slice links, and finally used supervised learning algorithms to predict future time links state. In 2019, Lei et al. [53] used GCN to learn the network topology features of each time slice, used LSTM (Long Short Term Memory) to capture the evolution pattern of mul-tiple continuous time slice dynamic networks, and then used GAN (Generative Adversar-ial Network) to generate a representation of links in future time slices. In 2017, Zhang et al. [54] proposed the WLNM link prediction model, which extracts the h-hop closed sub-graph of the target node pair and sends the adjacency matrix of the subgraph to the fully connected layer for learning, which improves the link prediction algorithm results. Link prediction methods based on graph representation learning are difficult to capture deeper network structural relationships in complex networks and more complex relationship features between nodes due to limited walk steps and aggregation methods, resulting in lower algorithm accuracy.”
Point 3: The considered reference data sets are adequate, but the authors create small subsets out of the original data sets, or at least this is what their presentation suggests. Given the relatively small number of included vertices and edges(network links), the authors should discuss on their proposed model suitability to scale up well for large real-world settings.
Response 3: Thank you very much for pointing it out. Our basic algorithm PLAS is composed of four steps:
- Extraction of subgraphs
- Graph labeling
- Subgraph encoding
- Fully connected layer learning
In the above four steps, our optimize the Graph labeling step by make promotion to WL(Weisfeiler Lehman) algorithm in only calcuating the shortest distance from nodes in the subgraph to destination node by skip the subgraph node label updating process in each iteration , resulting in less time complexity. And on other step such as Extraction of subgraphs, we set the samping node number threshold value to decreace algorithm time complexity. With the scale line up of dataset , the increase in time complexity of our algorithm is far below the n2 scale which is always the time complexity boundary of graph prcessing algorithms.
Point 4: The authors should also present the algorithm/methodology that was used in order to generate the experimental data sets.
Response 4: Thank you for pointing it out. All the Dataset belong to classicial complex network is coming from soma classicial link prediction papers . While USAir [15] is the transportation network of American Airlines, NS is the co-author network composed of researchers on the network [16], Pb is the blog network of the U.S. government [17], Yeast is the network composed of protein interactions in yeast [18], Cele is the divine network composed of neurons of a worm [19], power is the power network in the western United States [19], and router is the Internet network composed of routers [20]. Furthermore, we introduced the dataset Bupt5GMEC which is the real network structure of 5G MEC Access Network in BUPT while it demonstrates the inside routing structure of BUPT for 5G MEC access experimental environment through a MEC Hub Node . And this 5G MEC access experimental environment has be used to evaluate some routing links in 5G MEC experiments in Figure 3. Besides that, we also found some other important 5G MEC Access Network with the name CM1-5G and CM2-5G from other 5G MEC experiment joined developed with China Mobile. CM1-5G and CM2-5G are all real network dataset reflecting the real network structure of 5G Acess netwok collected by inner routers from the 5G project of China Mobile.
Finally, the real AS topology dataset ITDK0304S of Global Internet is added to Table 1 to carry out evaluation on the largest subset of real Internet topology [27]. ITDK0304S is generated and supplyed on Internet by other researcher and cut to the litte part by by using community detection algorithm FastGN with promotion to algorithm GN which is too time comsuming algorithm.
- Reference of FastGN:
Newman M E J. Fast algorithm for detecting community structure in networks [J]. Physical Review
E, 2004, 69.articleNO: 066133
- Reference of GN:
Newman M E J. Detecting community structure in networks [J]. European Physical Journal B, 2004, 38(2):321−330
Point 5: The hardware and software specifications of the experimental infrastructure and not clear, they should be fully presented.
Response 5: The hardware and software specification details of the experimental infrastructure have been put into Section 4 of the article as follows:
“Experiment environment: Stand-alone processor AMD Ryzen 7 4800H with Radeon Graphics 2.90 GHz, NVIDIA GeForce RTX 2060, memory 16G, operating system is Windows 10, the programming language is Python, based on the deep learning framework Pytorch.”
Point 6:The English language should be fully proofread and improved, eventually using the help of a native English speaker.
Response 6: Thank you for pointing it out and we have checked the Englsih grammar in the whole paper and try our best to promote it.
Round 2
Reviewer 2 Report
Manuscript is fine and suitable for publishing
Author Response
OK , thank you very much:)
Reviewer 4 Report
I acknowledge that the authors addressed some of my concerns. Nevertheless, they should re-check points 3 and 4 from my previous review report, as their responses are out of scope concerning these two points, especially concerning the presentation of the data sets generation methodology, and the appropriateness of generated data sests. Unfortunately, the authors have simply not clarified this aspect. Therefore, I am forced to provide a "Major" recommendation, in order to offer them the opportunity to address this problem.
The English is generally fine, while another round of proofreading would still be advised.
Author Response
Response to Reviewer Comments
Comments and Suggestions: I acknowledge that the authors addressed some of my concerns. Nevertheless, they should re-check points 3 and 4 from my previous review report, as their responses are out of scope concerning these two points, especially concerning the presentation of the data sets generation methodology, and the appropriateness of generated data sests. Unfortunately, the authors have simply not clarified this aspect. Therefore, I am forced to provide a "Major" recommendation, in order to offer them the opportunity to address this problem.
Response: Thank you very much for pointing it out. In the revised manuscript, we have made corresponding supplements and amendments to the point 3 and point 4 raised. And further more, we have checked the Englsih grammar in the whole paper and try our best to promote it.
- Point 3: The considered reference data sets are adequate, but the authors create small subsets out of the original data sets, or at least this is what their presentation suggests. Given the relatively small number of included vertices and edges(network links), the authors should discuss on their proposed model suitability to scale up well for large real-world settings.
Response for point 3, we explain the method adopted to construct the subset of the origin dataset, the reasons for adopting this approach, and the applicability of our algorithm to large-scale networks in the revised version. The newly added part is on line 417 of the article, we marked it in red, the details are as follows:
“For the construction of the ITDK0304S dataset, because the original dataset has some nodes separated from the overall network structure, there are few links between these nodes, and they are relatively independent of the overall network. As a result, it contains less information and research value and will affect model training. Therefore, we use the FastGN community detection algorithm to divide the original data set. We use several communities that can mainly represent the overall structure to construct the ITDK0304S data set and delete some individual communities with few nodes and nodes not worth researching. Since our model performs link prediction through extracted subgraphs, for large-scale real-world networks, we can adjust the hop count and threshold of sampling neighbors during subgraph extraction to ensure sufficient information for training without causing oversampling.”
Besides that, in optimizing the Graph labeling step by make promotion to WL(Weisfeiler Lehman) algorithm in only calcuating the shortest distance from nodes in the subgraph to destination node by skip the subgraph node label updating process in each iteration , resulting in less time complexity. And on other step such as Extraction of subgraphs, we set the samping node number threshold value to decreace algorithm time complexity. With the scale line up of dataset , the increase in time complexity of our algorithm is far below the n2 scale which is always the time complexity boundary of graph prcessing algorithms.
- Point 4:The authors should also present the algorithm/methodology that was used in order to generate the experimental data sets.
Response for point 4, we supplemented the details of the processing of the dataset in the revised version. The newly added part is on line 430 of the article, we marked it in red, the details are as follows:
“We abstract the real-world network structure data into the form of point-edge pairs. We take all the node pairs with edges in each network (as positive samples) and take the same number of node pairs without edges (as negative samples) to form a data set, and then randomly select 90% of the node pairs with and without edges as the training set and the remaining 10% of the data set as the test set. When we use the Node2Vec algorithm to calculate the implicit attributes of nodes, the node sequence obtained by random walk may imply information about whether there is a link between the target node pair, resulting in poor model generalization. Therefore, when the Node2Vec algorithm embeds nodes for each subgraph, links are added to the target nodes without link relationship so that the link information of any target node pair in the training set is the same.”
USAir ,NS, Pb, Yeast ,Cele ,router datasets are coming from their ogiginal papers.
Dataset Bupt5GMEC which is the real network structure of 5G MEC Access Network in BUPT while it demonstrates the inside routing structure of BUPT for 5G MEC access experimental environment through a MEC Hub Node .
CM1-5G and CM2-5G are from other innner 5G MEC experiment joined developed with China Mobile. CM1-5G and CM2-5G are all real network dataset reflecting the real network structure of 5G Acess netwok collected by inner routers from the 5G project of China Mobile and my email is shannondeng@bupt.edu.cn, I can send these dataset to by email.:)
The real AS topology dataset ITDK0304S is processed by FastGN algorithm from original AS topology dataset of Global Internet.